# Radiological and Clinical Efficacy of Intra-Arterial ^90^Y-DOTATATE in Patients with Unresectable, Progressive, Liver Dominant Neuroendocrine Neoplasms

**DOI:** 10.3390/jcm10081794

**Published:** 2021-04-20

**Authors:** Agnieszka Kolasińska-Ćwikła, Mirosław L. Nowicki, Artur J. Sankowski, Jakub M. Pałucki, John R. Buscombe, Lidia Glinka, Jarosław B. Ćwikła

**Affiliations:** 1Department of Oncology and Radiotherapy, Maria Skłodowska-Curie National Research Institute of Oncology, Wawelska 15, 02-034 Warsaw, Poland; jmpalucki@gmail.com; 2Department of Radiology, Hospital Ministry of Internal Affairs, Wołoska 137, 02-507 Warsaw, Poland; miroslnovicki@gmail.com (M.L.N.); art.san@gazeta.pl (A.J.S.); 3Department of Nuclear Medicine, Barts Health, West Smithfield, London EC1A 7BE, UK; jrb.wjnm@googlemail.com; 4Department of Cardiology and Internal Medicine, School of Medicine, University of Warmia and Mazury, Warszawska 30, 10-082 Olsztyn, Poland; lidka.glinka@gmail.com (L.G.); jbcwikla@interia.pl (J.B.Ć.); 5Diagnostic and Therapy Center—“Gammed” Lelechowska 5, 02-351 Warsaw, Poland

**Keywords:** neuroendocrine neoplasms, PRRT-intra-arterial (i.a.), ^90^Y-DOTATATE, radiological and clinical response, OS, PFS

## Abstract

This study was performed to determine if intra-arterial (i.a.) administration of ^90^Y DOTATATE can provide an effective and safe alternative to the accepted standard for i.v. of peptide receptor radionuclide therapy (PRRT) in liver-dominant metastases of gastrointestinal pancreatic neuroendocrine neoplasm (GEP-NEN). A single site, prospective, preliminary case series study included 39 patients with histologically proven liver-dominant NEN. PRRT in the form of 1.15GBq ^90^Y DOTATATE was given selectively into the liver via radiological catheterization of the hepatic artery, up to four times. The endpoint was radiological response (RECIST). Secondary endpoints assessed clinical well-being post-treatment, progression-free survival (PFS), overall survival (OS), and toxicity. Partial response (PR) was noted in 13% of subjects six weeks post-therapy, increasing to 24% at six months and dropping to 13% at 36 months. Disease progression (DP) was not seen at six weeks, was 5% at six months, and 47% at 36 months. Clinical response based on PS seen in 74% of patients at six weeks, 69% at six months, and 39% at 36 months had PFS and OS, respectively, of 22.7 months and 38.2 months. There was no difference in OS/PFS between those with RECIST PR and SD. One patient had significant toxicity (3%). Use of i.a. PRRT appears to be safe and effective in treating patients with liver-dominant NEN. In addition, the best OS (51 vs. 22 months) was seen when i.a. was used as an upfront treatment of bulky GEP-NEN liver metastases and not after i.v. ^90^Y DOTATATE. The use of i.a. ^90^Y DOTATATE PRRT appears to be safe and effective in treating patients with liver-dominant NEN.

## 1. Introduction

The therapeutic options for patients with progressive, advanced, metastatic neuroendocrine neoplasms of gastro-entero-pancreatic origin (GEP-NENs) with extensive liver disease are often limited [1]. In tumors of pancreatic origin, there are some systemic therapy trials, in particular mTOR blockers, tyrosine kinase inhibitors (TKI), or any other antiangiogenic drugs [2,3,4]. The RADIANT-4 study showed some good results using everolimus as a second-line therapy in progressive NET [5]. Peptide receptor radionuclide therapy (PRRT) has been available to patients for several years [6,7,8,9,10]. A recent randomized control trial, NETTER-1, demonstrated that good clinical results can be obtained using ^177^Lu-DOTATATE PRRT in progressive midgut tumors when compared to high-dose somatostatin analogs [11]. Therefore, ^177^Lu-DOTATATE is seen currently as the standard of care in progressive GEP-NEN and has been presented as such in international guidelines (ENETS, NANETS, and NCCN) [12,13,14]. For example, in the current ENETS guidelines, PRRT is the best therapeutic option in progressive somatostatin (SST) receptor-positive tumors, with homogenous SST receptor (SSTR) expression after the failure of somatostatin analogs including patients with extensive liver metastases [6,7,8,9,10,12,15].

PRRT allows precise binding of the therapeutic radionuclide, normally a beta emitter, to the cell via a high-affinity ligand such as DOTATATE, because its affinity for the SSTR sub-type 2 receptor is up to nine times greater than natural somatostatin [16,17]. This choice of DOTATATE was found to increase the efficacy of PRRT compared to the previously used DOTATAOC [18]. However, the radionuclide most commonly used in PRRT is ^177^Lu, which has a lower energy beta emission, resulting in a more limited tissue penetration range compared to the alternative ^90^Y, which has a highly energetic beta particle. This high energy of ^90^Y and its ability to penetrate up to 2 mm of tissue could be advantageous in the treatment of patients with large-volume liver metastases. The management of such patients is complex, and curative surgery can rarely be offered, as the metastases of GEP-NENs tend to occur diffusely through both liver lobes [1,2,3,6,12,13,14,15]. However, studies performed using systemically administered ^90^Y DOTATATE have shown a higher level of toxicity to the kidneys and some more significant bone marrow toxic effects than reported with ^177^Lu-DOTATATE [6,7,8,9,10,13,17,18,19,20].

An alternate strategy would be to deliver the ^90^Y DOTATATE directly to the liver via radiological cannulation of the hepatic artery and relying on the significant affinity of the GEP-NEN cells for the ^90^Y DOTATATE, resulting in a significant “first pass” effect and thus maximizing uptake into the GEP-NEN cells and reducing the systemic bioavailability of the ^90^Y DOTATATE and thus reducing the toxic effects to the kidneys and bone marrow but maximizing the radiation dose delivered to the metastases within the liver [15,21,22].

We and others have published some data on the efficacy of ^90^Y DOTATATE/DOTAoctreotide (DOTATOC/DOTA-Lanreotide (DOTA-LAN), and though we found benefit in survival and symptom control, it was more difficult to show a significant radiological response, especially using criteria such as RECIST, which depend primarily on contrast-enhanced CT or MRI [6,7,8,9,10,15,17,18,19,20,21,22,23,24]. The radiological response consistently under-predicted the clinical and survival benefits of the ^90^Y or ^177^Lu PRRT. The updated version of RECIST v. 1.1 can be less useful in patients with extensive liver disease, as only two rather than five liver lesions are measured [25,26]. In many studies, only a small number of patients had a significant measurable radiological response such as a partial response (PR), despite showing clear clinical benefit [6,7,8,9,10,11,12,27,28,29,30,31].

### The Aim of this Study

The primary aim of this study was to assess the possibility, in a preliminary study, of the possible efficacy of using two administrations of intra-arterial ^90^Y DOTATATE in patients with bulky unresectable liver metastases from GEP-NEN tumors. Treatment response was determined both radiologically, based on RECIST 1.0 criteria, which was felt to be appropriate for those with multiple liver lesions, and clinically, such as change in performance status—PS (ECOG), which in these patients is primarily related to symptom control. Further to this, there would be an assessment of both progression-free survival (PFS) and overall survival (OS) in patients treated with intra-arterial ^90^Y DOTATATE for extensive GEP-NEN liver metastases. Additional evaluation was performed of disease control rate (DCR) at selected time points in the patient’s clinical follow-up.

## 2. Materials and Methods

### 2.1. General

This was a prospective, open-label, interventional, single institution, preliminary case series study, which was approved by the Clinical Ethics Committee of the Central Clinical Hospital Ministry of Internal Affairs (reference 109/2004), where the study was conducted between May 2006 to December 2012 with a minimum six-year follow-up period. Prior to study inclusion, all patients understood the experimental nature of the treatment and all subjects signed their written informed consent form.

### 2.2. Patients

A total of 39 patients (Table 1), including 17 females (45%), with a mean age of 57.6 years (range 35–75 years) were recruited into the trial. The inclusion criteria include a histological diagnosis of gastro-entero-pancreatic neuroendocrine neoplasms (GEP-NEN): G1, G2, and in selected cases G3 (well-differentiated tumor cell with 20% < Ki-67 < 50%), [32] according to WHO and UICC/AJCC classification (TNM Classification of Malignant Tumors 8th Edition 2017). All patients had to have progressive disease noted within 12 months of entry into the study based on RECIST 1.0 criteria and unresectable liver metastases with at least 20% of liver involvement based on an increase in tumor size on CT or MRI. All patients had at least Krenning 2 uptake in the known liver lesions expressing somatostatin (SST) receptors, as seen on prior somatostatin receptor scintigraphy (SRS), using ^99m^Tc-[HYNIC, Tyr^3^]octreotide (TOC) (National Centre for Nuclear Research-Radioisotope CenterPolatom; NCNR; Otwock-Świerk, Poland) [33].

Though all patients had to have unresectable bulky liver metastases, they could also have extra-hepatic SRS avid disease. Biochemical progression was defined as increase serum CgA as ULN over 10% from baseline levels. Patients were recruited if they had bulky liver metastases that had failed to respond to intravenous prior PRRT (16 patients) or had such bulky liver metastases that it was thought optimal for their first line of PRRT therapy (Figure 1).

Exclusion criteria for entry into the trial were as follows: Hb < 80 g/L, WBC < 2 × 10^6^/L, platelets < 100 × 10^6^/L, creatinine level > 30 mg/L or GFR < 20 mL/min and poor performance status (PS), EOCG 3 or 4, and poorly differentiated NECG3 cancers. Contraindications also included pregnancy and known hypersensitivity to DOTATATE.

Previous treatments for the patient group included systemic i.v. ^90^Y DOTATATE PRRT in 16 patients (32%), and all but one patient had received long-acting somatostatin analog treatment, though this was suspended four weeks before the i.a. PRRT. Chemotherapy had previously been given to 12 (31%) patients, and local liver therapies such as selective trans-arterial embolization (TAE) and surgery had been performed on nine (23%) subjects (Table 2).

### 2.3. Therapy—Administration Protocol

An anti-emetic such as 8 mg Ondansetron (GlaxoSmithKlein, Brentford, Middlesex; UK) was administered i.v. 30 min, before the start of a 6 h i.v. infusion of 1 L of a lysine-containing amino acid solution such as Vamin 18 or Nephrotec, (Fresenius-Kabi, Bad Homburg; Germany). The amino acid infusion used to reduce any renal accumulation of the ^90^Y-DOTATATE was commenced 1.0–1.5 h before the PRRT.

The ^90^Y-DOTATATE was administered as a slow bolus injection of 10 mL over 20 min via an appropriately placed catheter or micro-catheter radiologically placed into the left or right hepatic artery. In patients with bi-lobar disease, the catheter would be placed and two infusions would be administered in the right and left hepatic arteries. Selective cannulation and infusion were performed on any aberrant hepatic arteries such as the accessory segment 4 artery.

Patients included in this analysis had at least two i.a. administrations of ^90^Y-DOTATATE with a mean activity of 1.15 GBq per session. The mean time between administrations of i.a.^90^Y-DOTATATE was 9.2 weeks (range 6–10 weeks); this primarily depended on the clinical condition of the patient and availability of angiography (Table 3).

### 2.4. Radiology

Any radiological response was measured via a standard multiphase contrast-enhanced CT scan with an abdomen arterial phase and a portal–venous phase of the chest, abdomen, and pelvis. The pre-treatment baseline CT had to have been performed within three months prior to the first i.a. ^90^Y-DOTATATE administration followed by six monthly intervals post-treatment. The initial arterial phase image was also used to help assess the hepatic vasculature pre-intervention. If CT was contraindicated, then MRI was performed with dynamic contrast enhancement (DCE) (Appendix A).

### 2.5. Biodistribution of the Radiotracer

Between 8 and 18 h post-therapy, the biodistribution of the ^90^Y-DOTATATE was determined using a “Bremsstrahlung” whole body scintigraphy and SPECT images using a dual-head gamma camera (e-cam; Siemens, Erlangen, Germany) equipped with medium energy collimation with a photopeak centered on 95 keV with a 50% window.

## 3. Assessment of Effectiveness

### 3.1. Clinical Response and Performance Status

An assessment of the patient’s general health and tumor-specific symptoms was performed by three observers (AKC, MLN, and JBC) before treatment, then at six weeks, and then three months after their last therapy episode. At each visit, the patient’s general health was assessed using the standard ECOG performance status (PS) scale.

The clinical symptoms of response on i.a. PRRT assessed in this study included appetite, malaise, weight change, the presence and intensity and frequency of abdominal pain, diarrhea, flushing, nausea, vomiting, fever, wheezing, and abdominal bloating. Analgesia and somatostatin analog requirements before and after treatment were recorded in clinical files.

To assess any biochemical or endocrine toxicity, serial measurements (pre-treatment, six weeks post-treatment, and then every three months) of plasma chromogranin-A (CgA) and if relevant 24-h urinary 5-hydroxyindole acetic acid (5-HIAA) and fasting gut hormones are performed for patients with biologically functioning pancreatic neuroendocrine tumors.

### 3.2. Image Analysis Radiological Response (RECIST v. 1.0)

Tumor response was determined by CT (or alternatively by MRI if CT was not possible) and scored, according to RECIST 1.0, by two independent radiologists with a special interest in GEP-NEN tumors (AJS and JMP) who were required to reach consensus on the size change of the five largest liver metastases seen on the pre-treatment liver scans. Any new liver metastases were also noted [34]. In addition, the diameters of all liver metastases greater than 10 mm were counted by each observer at each imaging time point to provide a determinate of total tumor liver metastatic load to determine the disease control rate (DCR).

### 3.3. Statistical Analysis and Patient Survival Analysis

Statistical analysis was performed using Statistica v.13.1 (TIBCO Software Inc. Palo Alto, CA; USA). Differences in performance status (PS) in the ECOG scale in patients before therapy and after completed radionuclide therapy were performed using Wilcoxon’s matched pairs test. Differences between data sets of two independent samples were performed using a Mann–Wilcoxon U-test. The overall survival (OS) and progression-free survival (PFS) were estimated for the total cohort of patients using the Kaplan–Meier estimator. Overall survival (OS) was defined as the time from the first administration of i.a. therapy until death from any cause or last follow-up as censored data. PFS was calculated from the start of i.a. PRRT to recorded disease progression according to RECIST or death. Comparison of OS and PFS between different groups of patients was assessed using the Cox–Mantel test and log-rank test. The proportions of patients who had a clinical response (PS status) and CT responses were calculated separately. *p* < 0.05 was considered statistically significant.

## 4. Results

Of the 39 treated GEP-NEN patients, 14 had tumors of pancreatic origin, 13 had tumors arising from the small bowel, four has tumors arising from the large bowel/rectum, and eight had metastatic NEN but with an unknown primary (CUP—cancer of unknown primary). There were 19 of these patients with secreting tumors, nine of whom had had previous i.v. PRRT. The most common histological stage was NETG2 occurring in 28 subjects (73%) compared to NETG1 in eight patients (22%) and NETG3 in two patients (5%). At the start of treatment, 29 patients (74%) had a PS of 1, eight patients (23%) had a PS of 2, and one patient had a PS of 0. Liver metastases of over 50% of total liver volume were seen in 13 patients (33%), liver metastases involving 25–50% of the liver was seen in 18 (46%), and a metastatic volume involving less than 25% of total liver volume was seen in eight subjects (21%) (Table 1). All patients had received at least one previous line of treatment (Table 2). Patients received a mean activity of 3.13GBq normally split between two treatment cycles (Table 3). The biodistribution of the ^90^Y-DOTATATE post-therapy in almost all cases perfectly matched the pre-treatment diagnostic ^99m^Tc-HYNICTOC imaging.

There was a significant clinical response as measured by the change in performance status (PS) after therapy (*p* < 0.01 Wilcoxon matched pair test). The PS improved in 29/39 patients (74%) at six weeks after PRRT, but in two patients, the PS had deteriorated. Of the 29 patients who had an improved PS at six weeks, 27 (93%) had maintained their improved PS; by 36 months, this improvement in PS was maintained in seven patients. The PS of four patients remained unchanged throughout the follow-up period

Based on RECIST criteria, at the six weeks post-imaging-based PRRT, no patient had suffered a DP, six months post-treatment the PR + SD (disease clinical response-DCR) was 95%, and of the 37 patients imaged at 12 months post-PRRT, the DCR was 92%. There were 15 patients still alive and imaged at 36 months after PRRT with a DCR of 53%. In those patients who had received both i.v.and i.a. PRRT, the DCR was 60% compared to just 40% in those receiving i.a. PRRT only. There were five patients with a PR at six weeks, but this had increased to nine at six months without further PRRT. At six months, two patients recorded SD. No patient had a CR (Table 4).

It was noted that 14 patients with PR during the follow-up had a mean tumor/liver ratio (T/L) of 0.28, compared to those patients who had only SD during follow-up with mean T/L = 0.48 (*p* < 0.002 Mann–Whitney U test). This reflected the higher tumor bulk in patients with SD than those with a PR. Four patients in whom i.a. was given without prior i.v. treatment with initial tumor shrinkage showed an increase in tumor size by 24 months (Figure 2); this was only seen in two patients who had had prior i.v.^90^Y-DOTATATE (Figure 3).

The median PFS and median OS for all patients were 24.1 months (CI 16.7–30.9) and 38.2 months (CI 34.0–71.2) (Figure 4).

There was a significantly different increase in median OS if patients were treated with i.a. ^90^Y-DOTATATE as their initial treatment at 52.1 months (CI 44.3–104.8) compared to those who had received i.v. ^90^Y-DOTATATE with a median OS of 22.2 months (CI 16.3–52.3) (*p* = 0.02); there was a borderline difference noted in PFS 12.3 months (CI 8.7–30.5) for those who had i.a. ^90^Y-DOTATATE compared to 28.4 months (CI 21.4–37.6) for those who received iv. ^90^Y-DOTATATE first (*p* = 0.056) (Figure 5).

In those patients with a DCR at six and 12 months, the survival advantage of treating bulky liver metastases with upfront i.a. ^90^Y DOTATATE was maintained. In those patients who had a DCR evaluated at six months, was 95%; at 12 months, it was 89%; and at 24 months in 30 alive patients, it was 73%. The median PFS (+/− 95% CI) at six, 12, and 24 months of DCR for all patients was, respectively, 23.9 months (18.0–32.9), 26.1 months (20.1–36.2), and 28.3 months (23.4–47.4) (Figure 6 and Figure 7).

The most common adverse event seen was Grade 1 anemia seen at six weeks post-therapy at a rate of 39%; in addition, 32% suffered a mild Grade 1 leukopenia, also peaking at six weeks post-PRRT (Table 5). There was a single patient who had significant anemia (Grade 3), which was managed conservatively. No other significant hematological or clinical biochemical or endocrine adverse events were recorded. In particular, there was no significant change in platelet count or adverse effect on liver function test results.

## 5. Discussion

The results of this study show that the use of i.a. ^90^Y DOTATATE could be an interesting strategy in which patients with bulky GEP-NEN metastases to the liver could be treated. It would appear from our preliminary findings in this small patient group that the best overall survival is obtained by the use of at least two cycles of i.a.1.1GBq ^90^Y DOTATATE is given upfront for such bulky disease within the liver with no apparent increase in toxicity when compared to i.v. ^90^Y DOTATATE.

Following the NETTER-1 trial, ^177^Lu DOTATATE has become the radionuclide treatment of choice in patients with metastatic GEP-NEN [11,12,13,14,35]. It has been shown to be both effective and safe. In a systematic review, the overall disease control rate (DCR) for ^177^Lu DOTATATE was 80%, and this was achieved without significant adverse events [36]. In the same systemic review, the DCR for i.v. ^90^Y DOTATATE was 92%. However, these results are not directly comparable, as the patient groups were probably not identical, and this improved efficacy was achieved with a rate of about 4% in significant adverse events (AEs).

The standard regime for systemic ^90^Y DOTATATE therapy was to give four cycles of 3.0-4.0 GBq, calculated to keep the renal dose below 23 Gy [20,29,35]. This, however, could lead to some significant bone marrow toxicity, with reports of late-onset myleodyspalstic syndrome (MDS) and leukemia [36]. There is no doubt that the extended path length of the ^90^Y beta could result in more tumor destruction, but this appeared to be handicapped by the increased adverse events rate [20,29,31,35,36]. It had been noted that liver metastases have historically been seen as a sign of a poor prognosis in GEP-NEN tumors, and untreated, they can be the cause of premature death either by the release of related hormones, loss of liver synthetic function, or pressure effects on other vital organs [37].

Therefore, it was logical to use techniques allowing local delivery of the ^90^Y DOTATATE to the liver. There had been some attempts to use particulates such as ^90^Y resin and glass to treat liver metastases in neuroendocrine tumors, but this can lead to post-treatment embolic syndrome, and its use has tended to be limited to tumors that do not express somatostatin receptors [38,39]. Comparison with purely embolic treatment such as trans-arterial embolization (TAE) and trans-arterial chemo-embolization (TACE) is more problematic, as neither of these techniques has the compounding factor of the use of high-energy beta-radiation within the liver that is seen with ^90^Y labeled particulates or DOTATATE.

Early reports did suggest that giving the ^90^Y labeled somatostatin analogs via an intra-arterial catheter in the hepatic artery can improve the response rate in liver metastases. The first reports, however, either used a regime with ^90^Y DOTATATE but given both systemically and intra-arterially or used a different peptide such as ^90^Y Lanreotide [15,22]. The largest series reporting the use of i.a. PRRT was that in a group of 23 patients receiving 1-2 cycles of just 1 GBq ^90^Y Lanreotide, a 79% DCR was achieved with a median overall survival (OS) of 15 months. There was no significant toxicity in this group of patients, the majority of whom had metastases representing greater than 50% of their total liver volume [15]. Lanreotide, however, has a lower affinity to GEP-NEN tumors compared with DOTATATE, so a DOTATATE-based approach may be still more efficacious [16,17,18].

Previously reported use of systemic ^90^Y DOTATATE in a similar group of patients from the same institutions as the patients treated in this study in which up to four cycles of 3.7 GBq each was administered showed an 87% DCR at six months post last therapy very similar to the 92% achieved in patients with liver metastases treated with at least two sessions of i.a. injection of 1.15 GBq each of ^90^Y DOTATATE in the current study. The mean cumulative activity administered in this study was only 3.13 GBq for all patients and was lower in those with previous i.v. PRRT vs. only i.a. 2.8 vs. 3.4 GBq, respectively. In our previous study, the mean administered activity in i.v. therapy was 11.2 GBq, which is almost four times greater than in subjects in the current trial who had initially (only) i.a. PRRT [9].

With the systemic administration of ^90^Y DOTATATE, the median PFS was 17 months (CI 16.4-21.2) compared to 24.1 months (CI 16.7-30.9) with the current i.a. ^90^Y PRRT. The median OS for those treated with systemic ^90^Y DOTATATE was 22 months (CI 20.4-26.7) compared to 38.2 months (CI 34.0-71.2) for those receiving i.a. ^90^Y DOTATATE [9]. Our study indicates additional, significant differences in median PFS and OS between when the i.a. approach of PRRT was given first; this may be due to the ability to deliver a higher therapeutic dose of the radionuclide to the liver metastases with reduced systemic side effects.

Another potential explanation of our results in both groups of patients who had current i.a. therapy was the selection of more aggressive tumor cell lines surviving treatment by previous i.v. PRRT in those patients who then relapsed and were treated with further i.a. PRRT. This phenomenon probably related to the natural history of the development of GEP-NEN, with the survival of more aggressive clones of cancer cells after anticancer therapy, like PRRT, meaning those clones of cells that survive are less sensitive to second-time beta irradiation [35].

The lack of radiological PR has been noted before in GEP-NEN patients treated with radioisotopic therapy in many others reports [7,8,9,10,15,17,18], and the results of this study are similar to a large series of over 300 patients from Rotterdam treated with ^177^Lu DOTATATE, where both partial response (PR) and disease stability (SD) as assessed by radiological criteria were related to a good overall prognosis [19]. In the NETTER1 randomized controlled trial, a radiological response was seen in 18% of patients, which is probably not significantly different from the 24% seen in the patients treated with i.v. ^90^Y DOTATATE in our previous reports [9,11]. The improved DCR at 36 months was seen in both those patients who had received previous systemic PRRT and those who were treated upfront with i.a. PRRT, indicating that i.a.^90^Y DOTATATE could be used safely after previous systemic PRRT, but that sequential treatment may not be the ideal scenario in this difficult group of subjects, and upfront i.a. ^90^Yttrium DOTATATE may be the best way forward in those patients with bulky liver metastases from GEP-NEN tumors.

In this trial, we noted a low rate of any AEs compared to the group of patients who were treated previously using i.v.^90^Y DOTATATE, as the reported Grade 3 and four toxicity (AEs) rate for those receiving systemic ^90^Y DOTATATE was 10% compared to just 3% in those treated with i.a. ^90^Y DOTATATE. As the patients treated by these two studies may have a number of different characteristics, the primary one being the liver-dominant disease in the i.a. group, it can be noted that it is unlikely that the i.a. ^90^Y DOTATATE is less effective and more toxic than the use of systemic ^90^Y DOTATATE. It is useful to note the Grade 3 and 4 hematological toxicity was 9% in those treated with ^177^Lu DOTATATE on the NETTER1, which is higher than seen in our group of patients treated with i.a. ^90^Y DOTATATE [11,19]. Though it is not possible to make a direct comparison between the two groups, it does suggest that low activity i.a. ^90^Y DOTATATE is as effective and certainly no more toxic than the standard four cycles of 7.4GBq ^177^Lu DOTATATE. However, the small number of patients within our study group means that further work in this area in a bigger patient group would be needed to confirm these findings.

## 6. Conclusions

The conclusion of this preliminary study is that treatment with two cycles of 1.1 GBq i.a. ^90^Y DOTATATE six weeks apart may be as effective and safe as the recommended cumulative activity of 28 GBq of ^177^Lu DOTATATE or 14 GBq of ^90^Y DOTATATE given over i.v. at eight months in those patients with bulky liver GEP-NEN metastases, with some evidence of improvement in overall response rate, including disease control rate (DCR), clinical response (PS), and also PFS and OS when given as upfront i.a.^90^Y DOTATATE. A larger multi-center trial should now be considered to determine if such an approach should be considered as an alternative or adjunctive form of PRRT in those with liver-dominant unresectable GEP-NEN metastases in their liver.

## Figures and Tables

**Figure 1 jcm-10-01794-f001:**
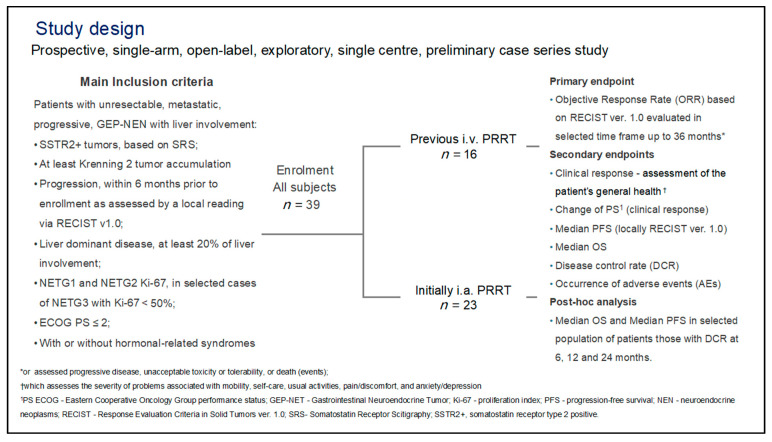
Study design, including main inclusion criteria, selection of two groups of patients, and endpoints.

**Figure 2 jcm-10-01794-f002:**
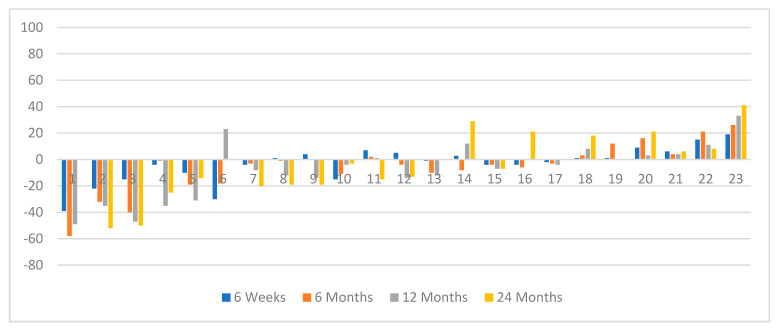
The waterfall plot of ORR in selected time frame points of evaluation at six weeks, six months, 12 months, and 24 months in the group with initially i.a.PRRT (*n* = 23).

**Figure 3 jcm-10-01794-f003:**
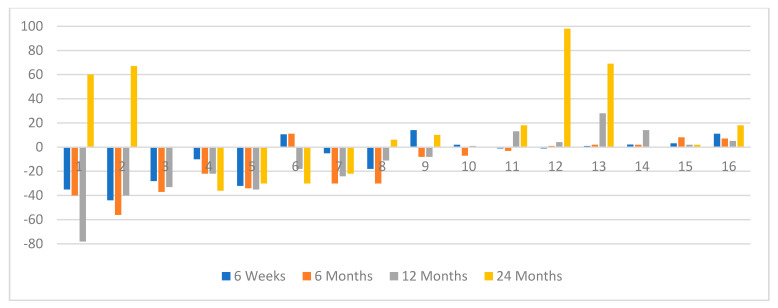
The waterfall plot of ORR in selected time frame points of evaluation at six weeks, six months, 12 months, and 24 months in the group with previous i.v. PRRT (*n* = 16).

**Figure 4 jcm-10-01794-f004:**
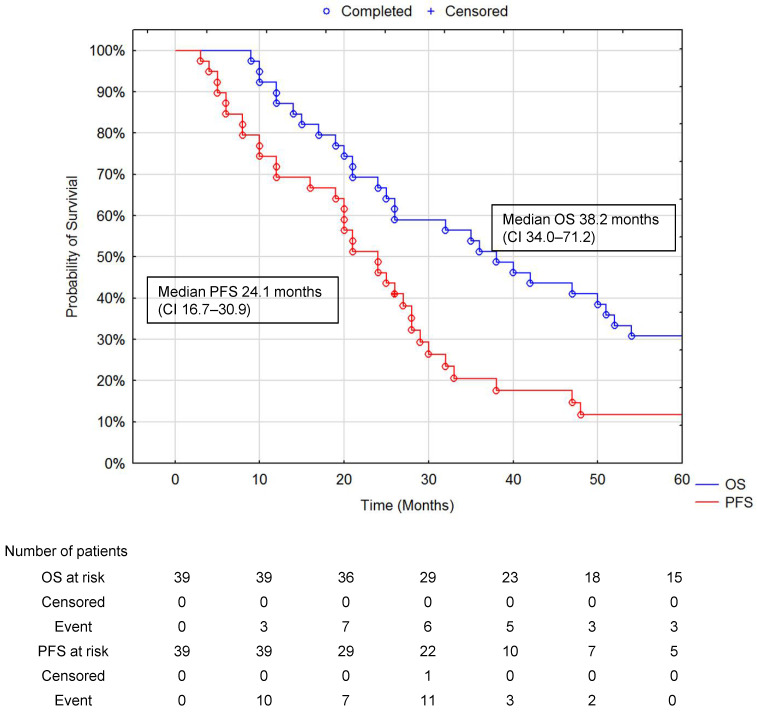
OS and PFS in all patients during follow-up. Median OS 38.2 months (95% CI 34.0–71.2) and median PFS 24.1 months (95% CI 16.7–30.9). Data are presented for all patients who received at least two doses of PRRT during the study. The number of subjects remaining at risk below is 10% of cases in any group.

**Figure 5 jcm-10-01794-f005:**
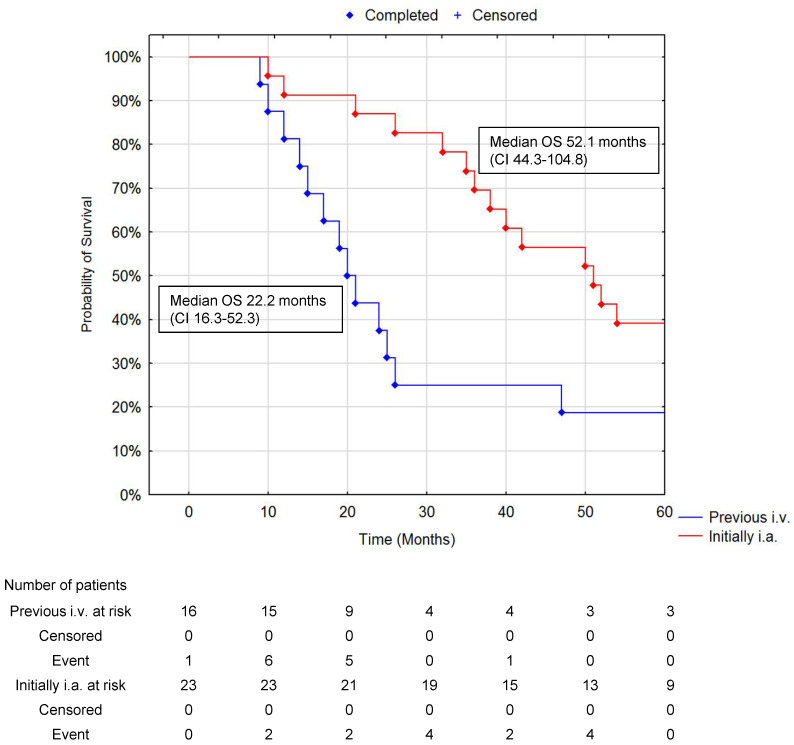
Comparison of OS in patients with previous i.v. PRRT and initially i.a. PRRT median 22.2 (CI 16.3–52.3) vs. 52.1 (CI 44.3–104.8) (*p* = 0.02 Cox Mantel Test). The number of subjects remaining at risk is below 10% of cases in any group.

**Figure 6 jcm-10-01794-f006:**
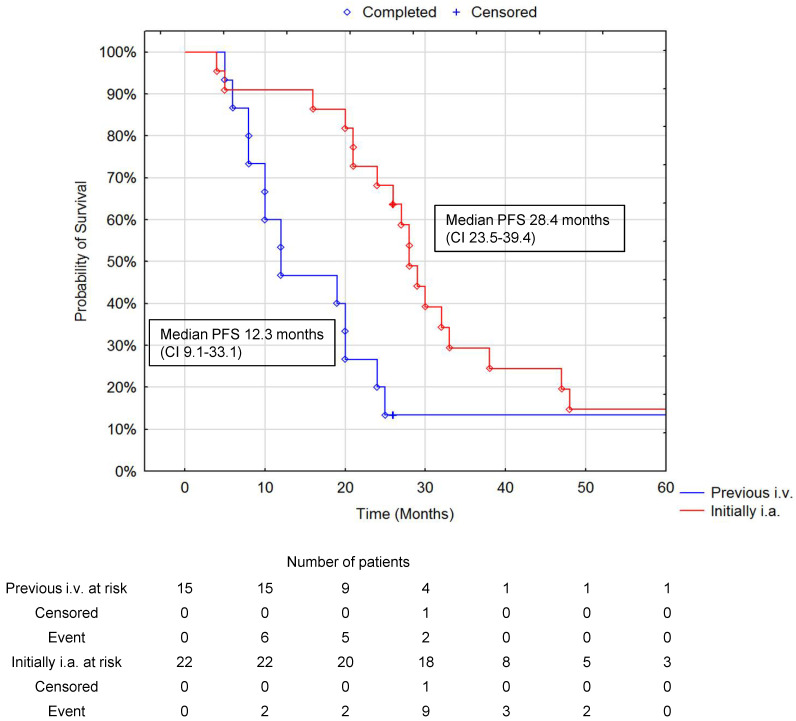
Comparison of PFS in patients with previous i.v. PRRT at 12.3 months (95% CI 9.1–33.1) and initially i.a. PRRT at 28.4 months (95% CI 23.5–39.4) at six months of DCR. (*p* = 0.048 Cox–Mantel test). Disease control rate was defined as the proportion of patients with partial response (PR) or stable disease (SD), not patients with complete response (CR). The number of subjects remaining at risk was below 10% of cases in any group.

**Figure 7 jcm-10-01794-f007:**
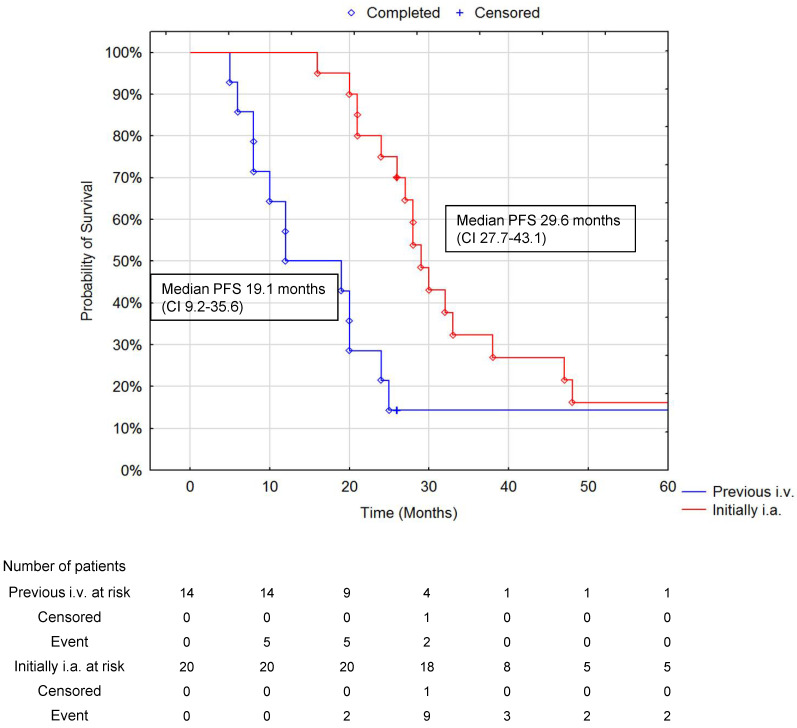
Comparison of PFS in patients with previous i.v. PRRT at 19.1 months (95% CI 9.2–35.6) and initially i.a. PRRT at 29.6 months (95% CI 27.7–43.1) at 12 months of DCR (*p* = 0.026 Cox–Mantel test). Disease control rate was defined as the proportion of patients with partial response (PR) or stable disease (SD), not patients with complete response (CR). The number of subjects remaining at risk was below 10% of cases in any group.

**Table 1 jcm-10-01794-t001:** Clinical data of all patients and those with previous i.v. PRRT and native i.a. PRRT.

	All *n* = 39	Previous i.v. PRRT *n* = 16 (%)	Only i.a. *n* = 23 (%)
**Male to female**	22/17	12/4	12/11
**Age in years** (mean, SD)	56.4 (9.2)	55.4 (8.2)	57.5 (9.9)
**Primary site NEN origin**			
Pancreas	14 (36)	7 (44)	7 (30)
Small Bowel	13 (33)	5 (31)	8 (35)
Large Bowel/Rectum	4 (10)	1 (6)	3 (13)
CUP (Cancer of Unknown Primary)	8 (21)	3 (19)	5 (22)
**Secretor tumors**	19 (49)	9 (56)	10 (43)
**Tumor grade**			
NETG1 (Ki-67 ≤ 2%)	8 (22)	4 (27)	4 (17)
NEG2 (2 < Ki-67 ≤ 20%)	28 (73)	10 (67)	18 (78)
NETG3 (20% < Ki-67 < 50%)	2 (5)	1 (6)	1 (5)
**Hepatic load (tu_Volume_/liver_Volume_)**	0.41%	45%	38%
≤25%	8 (21)	3 (19)	5 (22)
< 25% x ≤ 50%	18 (46)	6 (38)	12 (52)
>50%	13 (33)	7 (43)	6 (26)
**WHO Performance status (PS) initial**	All *n* = 39	Previous iv. PRRT *n* = 16 (%)	Only i.a. *n* = 23 (%)
0—normal activity	1 (3)		1 (5)
1—restricted activity	29 (74)	11 (69)	18 (78)
2—in bed ≤ 50% of the time	9 (23)	5 (31)	4 (17)
**Initial basic hematology and kidney creatinine level**	All *n* = 39	Previous iv. PRRT *n* = 16	Only i.a. *n* = 23 mean value
WBC (×10^9^/L)	6.78	7.34	6.37
Hb (g/dL)	12.8	12.0	13.4
Platelet (×10^6^/L)	288	297	282
Creatinine level mg/dL	0.97	1.24	0.78
CgA x ULN; mean (95% CI)	34.5 (19.8–49.2)	48.8 (11.6–81.9)	27.6 (13.9–41.3)

Ki-67 thresholds as per WHO and UICC/AJCC classification 2017 classification as NETG1 (Ki-67 < 3%), NETG2 (2% < Ki-67 ≤ 20%), and NETG3 (Ki-67 > 20% but below 50%). CgA—chromogranin A.

**Table 2 jcm-10-01794-t002:** Previous systemic therapies, considering all patients and those with previous i.v. and only i.a. PRRT.

Therapy Approach before i.a. PRRT	All *n* = 39 (%)	Previous iv. PRRT *n* = 16 (%)	Only i.a. PRRT *n* = 23 (%)
Analogs SST (long-acting)	32 (82)	13 (81)	19 (83)
Chemotherapy any type	12 (31)	9 (56)	3 (13)
Previous i.v. PRRT	16 (41)	16 (100)	
Local Liver therapy TA, surgery	9 (23)	4 (25)	5 (22)

**Table 3 jcm-10-01794-t003:** PRRT therapy scheme, consider all patients previous i.v. PRRT and native i.a. PRRT.

Parameter	All Subjects *n* = 39	Previous i.v. PRRT *n* = 16 (%)	Native i.a. *n* = 23 (%)
Mean activity (GBq of ^90^Y DOTATATE) per session	1.15	1.12	1.16
Mean Cumulative activity (GBq of ^90^Y) per therapy (range)	3.13 (1.4–4.1)	2.8 (2.0–4.1)	3.4 (1.4–4.1)
Mean (range) time between each therapy sessions (weeks)	9.2 (6–12)	10 (8–12)	8.4 (6–10)

**Table 4 jcm-10-01794-t004:** Objective response rate (ORR) and clinical response after six weeks (6 W) and then after six months (6 M), 12 months (12 M), 24 months (24 M), and 36 months of follow-up for all patients after i.a.PRRT (*n* = 39). Additional data sets in selected populations of patients after previous i.v. PRRT and in those with initially i.a. PRRT.

Response	RECIST 6W, *n* (%)	Clinical Response 6 W	RECIST; 6 M *n* (%)	Clinical Response 6 M	RECIST 12 M, *n* (%)	Clinical Response 12 M	RECIST 24 M *n* (%)	Clinical Response 24 M	RECIST 36 M *n* (%)	Clinical Response 36 M
All subjects *n* = 39										
PR	5 (13)	29 (74)	9 (24)	27(70)	9 (24)	24 (63)	5 (16)	15 (46)	2 (13)	7 (39)
SD	34 (87)	8 (21)	27 (71)	6 (15)	25 (68)	4 (11)	18 (58)	6 (18)	6 (40)	4 (22)
DP		2 (5)	2 (5)	6 (15)	3 (8)	10 (26)	8 (26)	12 (36)	7 (47)	7 (39)
Previous i.v. PRRT *n* = 16										
PR	3 (19)	12 (75)	6 (38)	10 (63)	4 (25)	7 (44)	3 (23)	3 (23)	2 (40)	2 (40)
SD	13 (81)	3 (19)	10 (62)	4 (25)	11 (69)	2 (12)	6 (46)	1 (8)		
DP		1 (6)		2 (12)	1 (6)	7 (44)	4 (31)	9 (69)	3 (60)	3 (60)
Initially i.a. PRRT *n* = 23										
PR	2 (9)	17 (74)	3 (14)	17 (74)	5 (23)	17 (77)	2 (11)	12 (23)		5 (38)
SD	21 (91)	5 (22)	17 (77)	2 (9)	14 (67)	2 (9)	12 (67)	5 (8)	6 (60)	4 (31)
		1 (4)	2 (9)	4 (17)	2 (10)	3 (14)	4 (22)	3 (69)	4 (40)	4 (31)

M-Months; W-Weeks.

**Table 5 jcm-10-01794-t005:** Treatment-related hematological and kidney adverse events (AEs) in the whole group of treated patients, percentages in brackets.

	Initial Week 0 *n* = 38	Week 6 after PRRT, *n* = 38	6 Months *n* = 35	12 Months *n* = 29	24 Months *n* = 20
**WBC**					
Grade 1	5 (13)	12 (32)	9 (26)	7 (24)	1 (6)
Grade 2			1 (3)	1 (3)	
Grade 3					1 (6)
**Hb**					
Grade 1	10 (26)	15 (39)	6 (21)	6 (21)	3 (15)
Grade 2	2 (5)	2 (5)	2 (7)	2 (7)	1 (5)
Grade 3		1 (3)	1 (3)	1 (3)	
**Creatinine**					
Grade 1	2 (5)		2 (6)	4 (14)	
Grade 2	2 (5)	3 (8)	2 (6)	2 (7)	
Grade 3					

## Data Availability

This was single site, prospective, preliminary case series.

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
