# Peer review of "Radiological and Clinical Efficacy of Intra-Arterial 90Y-DOTATATE in Patients with Unresectable, Progressive, Liver Dominant Neuroendocrine Neoplasms"

_jcm, 2021, doi:10.3390/jcm10081794_

Round 1

Reviewer 1 Report

Firstly, I would like to congratulate the authors on their work on a rare disease. As medical knowledge progresses, data on neuroendocrine tumours remains scarce. 

The authors performed a single site prospective open-label phase II study on administrating i.a. 90Y-DOTATATE in 39 patients with liver dominant NEN. The primary endpoint was radiological response while secondary endpoints were PFS, OS and toxicity. 

Based on the results of their study, the authors conclude that i.a. administration of up to four 1.1GBq i.a. 90Y DOTATATE is as effective and safe as the recommended cumulative activity of 28GBq 177Lu DOTATATE over 8 months and shows no inferiority concerning the chosen endpoints. 

While the idea behind the study is quite relevant and novel, I do have a few concerns and remarks I would like to state:

1) The study design does not seem appropriate to meet the study's aim. Comparing a group of patients with upfront i.a. administration of DOTATATE with patients with prior i.v. administration of PRRT and then i.a. DOTATATE does not seem suitable as these groups do differ substantially in duration and progression of disease. 

2) A case number of 39 seems quite low to give a statement on toxicity

3) The authors claim that this study is part of NCT04029428. However, NCT04029428 study protocol does not mention this work and therefore should not be associated to it. It rather seems like the authors performed a retrospective case series. 

4) The manuscript seems somehow incoherent. While some parts are written in perfect English, others would definitely benefit from a revision by a native speaker. There are no major mistakes but it seems like the manuscript's parts were written by different authors. 

To conclude, this study should be used as a pilot study or preliminary case series that a larger study or even RCT could be based upon. However, I find calling it a single site prospective open-label phase II study slightly discomforting. 

Author Response

  • The study design does not seem appropriate to meet the study's aim. Comparing a group of patients with upfront i.a. administration of DOTATATE with patients with prior i.v. administration of PRRT and then i.a. DOTATATE does not seem suitable as these groups do differ substantially in duration and progression of disease. 

We felt it was important to look at both groups of patients and it did indeed seem there was a difference in those patient’s survivial with the i.a 90Y DOTATATE was given as the first PRRT than when it was administered after prior i.v PRRT. We have discussed these differences nore fully and noted this is a preliminary case series study and further work in this area is needed.

  • A case number of 39 seems quite low to give a statement on toxicity

Thank you, this is better acknowledged and we have tried ot ensure our results are presented and discussed in a way that better reflects the small patient numbers studied and this cannot be seen as a definitive study

  • The authors claim that this study is part of NCT04029428. However, NCT04029428 study protocol does not mention this work and therefore should not be associated to it. It rather seems like the authors performed a retrospective case series. 

This was an error on our part and has been corrected though data was collected prospectively and not retrospectively this has been clarified in the methods section.

  • The manuscript seems somehow incoherent. While some parts are written in perfect English, others would definitely benefit from a revision by a native speaker. There are no major mistakes but it seems like the manuscript's parts were written by different authors. 

Thank you we have attempted to repedit to help it flow more as a single document with overall revision by a native English speaker

  • To conclude, this study should be used as a pilot study or preliminary case series that a larger study or even RCT could be based upon. However, I find calling it a single site prospective open-label phase II study slightly discomforting. 

Understood and we hope our changes make the reviewer feel less uncomfortable.

Reviewer 2 Report

In this single site, prospective, open label, phase II study the Authors assess the radiological response and survival outcomes of NEN patients following i.a. PRRT treatment.. In my view, the manuscript is of interest to the clinical NEN community and summarizes well valuable information; therefore I believe it would be of interest to the readership of JCM. However, there are some flaws that need to be addressed. The manuscript would benefit from further revision and clarification with respect to its limitations.

Reviewer's comments

  1. Please use standard nomenclature folowing the STROBE guideline to report the study results. In the PFS and OS analysis figure provide patient at risk numbers under the figure at different time points on the x axis. Please define a minimum number of subjects remaining at risk (e.g 10% of cases) after which Kaplan-Meier survival plots for time-to-event outcomes should be curtailed, as, once the number remaining at risk drops below this minimum, the survival estimates is no longer meaningful in the context of the investigation.
  2. Page 1, Introduction: Please use consistently the term NEN  instead of NET.
  3. Page 1, Introduction: ”In tumors of pancreatic origin there are some chemotherapy trials in particular mTOR blockers, tyrosine kinase inhibitors (TKI) or antiangiogenic drugs”. mTOR inhibitors and tyrosine kinase inhibitors (TKI) are not chemotherapeutics. Tyrosine kinase inhibitors (TKI) include antiangiogenic drugs. Therefore, please consider rewriting this part.
  1. Page 4, Materials and Methods, Patients: Please clarify the grading classification system that the Authors applied and provide relevant reference. The term moderate differentiated tumor cell with 20%<Ki67<55% is not valid. Please specify whether only well-differentiated tumors were included in the study or also NECs. Please revise table 1 and the first paragraph of the Results section accordingly.
  2. Page 5, Materials and Methods, 3.2. Image analysis radiological response (RECIST v. 1.0). Please prvide reference for RECIST 1.0
  1. In Table 1 and also the first paragraph of the Results section the Authors apply the distinction foregut, midgut and hindgutt hat is no longer used in contemporary litterature. Please clarify the primary origin of the tumors included in this study instead.
  2. Please provide comparison ORR, DCR and clinical response between patients pretreated with i.v. PRRT and the ones only receiving i.a. PRRT. With respect to log-rank PFS analysis please define baseline for survival estimates (PRRT initiation?).
  3. Tables 5 and 6 could be omitted as this information is also given in the text and in the respective figures (5 and 6).
  4. Please consider providing data analysis on treatment outcomes (ORR, DCR and PFS) stratified by primary site, tumor grade and liver tumor burden.
  5. Please explain that QoL data are not provided in this study as in the Methods section the Authors described a protocoll including collection of QoL questionnaires.
  6. Please elaborate briefly on the main findings of the study in the first paragraph of the discussion section.
  7. Page 12, Discussion section: The Authors state: ”It had been noted that liver metastases are a sign of poor prognosis in GEP-NEN tumors and untreated often lead to premature death”. Please consider rephrasing as this statement is not completely correct. Depending on the primary site, GEP-NENs have a variable prognosis and often a long expected survival even in cases of distant stage disease, e.g. small intestinal NENs.
  8. Please briefly discuss in the discussion section alternative i.a. methods of embolization including TAE, TACE and radioembolization (SIRTEX) for liver dominant disease and address their efficacy.
  9. Please provide a paragraph in the end of the Discussion section with the limitations of the study, including the study design (lack of randomization, no control group) evolving classisfication systems as well as evolving treatment modalities during and after the study period, small sample size etc.

Author Response

  1. Please use standard nomenclature following the STROBE guideline to report the study results. In the PFS and OS analysis figure provide patient at risk numbers under the figure at different time points on the x axis. Please define a minimum number of subjects remaining at risk (e.g 10% of cases) after which Kaplan-Meier survival plots for time-to-event outcomes should be curtailed, as, once the number remaining at risk drops below this minimum, the survival estimates is no longer meaningful in the context of the investigation.

Thank you very much for pointing out this aspect, it has been corrected in the tables below Fig. K-M. The K-M curves have been corrected and adjusted for the length of follow-up time presented in the tables below taking into account The number of subjects remaining at risk below10% of cases at any group. All graphs and accompanying figures below corrected, applies to fig.4, 5, 6 and 7.

  1. Page 1, Introduction: Please use consistently the term NEN instead of NET.

Correction made, in all names NEN was used instead of NET.

  1. Page 1, Introduction: ”In tumors of pancreatic origin there are some chemotherapy trials in particular mTOR blockers, tyrosine kinase inhibitors (TKI) or antiangiogenic drugs”. mTOR inhibitors and tyrosine kinase inhibitors (TKI) are not chemotherapeutics. Tyrosine kinase inhibitors (TKI) include antiangiogenic drugs. Therefore, please consider rewriting this part.

Correction was made to the naming of therapeutic regimens in NEN of pancreas as follows:

In tumors of pancreatic origin there are some systemic therapy trials in particular mTOR blockers, tyrosine kinase inhibitors (TKI) or any other antiangiogenic drugs.

  1. Page 4, Materials and Methods, Patients: Please clarify the grading classification system that the Authors applied and provide relevant reference. The term moderate differentiated tumor cell with 20%<Ki67<55% is not valid. Please specify whether only well-differentiated tumors were included in the study or also NECs. Please revise table 1 and the first paragraph of the Results section accordingly.

The system that has been adopted is WHO and UICC/AJCC classification (TNM Classification of Malignant Tumors 8th Edition 2017). A literature item on NEN G3 defining the NETG3 group was added. NECG3 cases were excluded from the study, noted in the exclusion criteria.

Table corrected foregut as pancreas, midgut as small bowel; hindgut as large bowel/rectum.

Additionally, NECG3 was corrected to NETG3 in the table.

In the footer description of the table, corrected:

WHO and UICC/AJCC classification 2017;

NETG3 Ki-67>20% but below 50%.

  1. Page 5, Materials and Methods, 3.2. Image analysis radiological response (RECIST v. 1.0). Please prvide reference for RECIST 1.0

A RECIST 1.0 literature item has been added as follows:

No of reference: 34. Therasse P, Arbuck SG, Eisenhauer AE, Wanders E, Kaplan J, Rubinstein R, Verweij L, Van Glabbeke J, van Oosterom M, Christian A, Gwyther, S. New guidelines to evaluate the response to treatment in solid tumors. Journal of the National Cancer Institute, 2000;92, 205-216.

  1. In Table 1 and also the first paragraph of the Results section the Authors apply the distinction foregut, midgut and hindgutt hat is no longer used in contemporary litterature. Please clarify the primary origin of the tumors included in this study instead.

A correction of the nomenclature was made by using modern nomenclature operating on organs. The foregut group includes NEN of pancreas, midgut - NEN of small intestine and hindgut - NEN of large intestine including rectum.

  1. Please provide comparison ORR, DCR and clinical response between patients pretreated with i.v. PRRT and the ones only receiving i.a. PRRT. With respect to log-rank PFS analysis please define baseline for survival estimates (PRRT initiation?).

Comparative analysis of ORR, DCR and clinical response are shown in Table 4. For all patients and for patients in both the i.v. and i.a-only groups, the starting point defining both PFS and OS was the first administration of i.a. PRRT. Note included in Materials and Methods in the Statistical Analysis section. Because of the sample size, we used the Cox-Mantel test for comparative analysis of potential differences between groups in median OS and PFS and additionally checked the data using the log-rank test. Due to the size of the study groups, we only presented data from the Cox-Mantel test in the P evaluation.

  1. Tables 5 and 6 could be omitted as this information is also given in the text and in the respective figures (5 and 6).

Table 5 and 6 removed, information from the table included in the text regarding the treatment of the whole group of patients and additionally in each study group in Fig. 6 and Fig. 7, respectively.

  1. Please consider providing data analysis on treatment outcomes (ORR, DCR and PFS) stratified by primary site, tumor grade and liver tumor burden.

Due to the size of the study group, it seems that the evaluation of treatment outcomes according to primary location, distinguishing pancreatic tumors n=14, small bowel n=13 colon/rectum n=4 and CUP n=8 may be subject to high error due to the low size of the individual patient groups mentioned above. Another element affecting the accuracy of results may be the division into patients previously treated with i.v. PRRT n=16 and those treated only with i.a. PRRT n=23. Another element differentiating the study group is the degree of liver involvement and overall stage of the disease process with the presence of metastatic and/or primary lesions, which further leads to the identification of potential further subgroups of subjects.

 Below, for example, we present ORR and DCR after 6 weeks, 6 mo, 12 mo and 24 mo in the group of patients with NEN of pancreas and NEN of small bowel. Presented as a table below.

Overall, a very similar response to treatment defined as DCR in both groups of treated patients those with small bowel and those with pancreatic NETs is marked. In the pancreatic NET group, PR is more frequently observed especially in the evaluation after 6 mo, 12 mo and 24 mo as included in the table. In addition, a similar rate of nonresponse in both small bowel and pancreas groups in ORR assessment as DP is evident.

It appears that generating a K-M estimator to calculate median PFS in such small numbers of patients and comparing them may not be meaningful. The number of subgroups will generate too much error for any potential differences in PFS, between groups. Therefore, despite the attractiveness of performing such an analysis, we wish to stop at evaluating only the overall data and after dividing into patients with prior i.v. PRRT and those with i.a. PRRT treatment.

ORR

DCR

6 weeks

6 Mo 

12 mo

24 mo

6 weeks

6 Mo

12 mo

24 mo

Pancreas n=14

N=14

N=14

N=13

N=11

N=14

N=14

N=13

N=11

PR

3 (21)

5 (36)

5 (38)

4 (36)

PR+SD

14 (100)

14 (100)

12 (92)

8 (73)

SD

11 (79)

9 (64)

7 (54)

4 (36)

DP

1 (8)

3 (27)

DP

1 (8)

3 (27)

Small Bowel n=13

N=13

N=13

N=13

N=11

N=13

N=13

N=13

N=11

PR

1 (8)

1 (8)

1 (9)

PR+SD

13 (100)

12 (92)

12 (92)

8 (73)

SD

13 (100)

11 (84)

11 (84)

7 (64)

DP

1 (8)

1 (8)

3(27)

DP

1 (8)

1 (8)

3 (27)

  1. Please explain that QoL data are not provided in this study as in the Methods section the Authors described a protocol including collection of QoL questionnaires.

We apologize for including information regarding the assessment of quality of life based on EORTC questionnaires in the Materials and methods section. The analysis on quality of life is the subject of a separate study, where we are trying to collect data from different types of systemic treatment for advanced NENs. The paragraph on QoL is removed.

11.Please elaborate briefly on the main findings of the study in the first paragraph of the discussion section.

Thank you we have now done this

  1. Page 12, Discussion section: The Authors state: ”It had been noted that liver metastases are a sign of poor prognosis in GEP-NEN tumors and untreated often lead to premature death”. Please consider rephrasing as this statement is not completely correct. Depending on the primary site, GEP-NENs have a variable prognosis and often a long expected survival even in cases of distant stage disease, e.g. small intestinal NENs.

Agreed but overall liver metastases can be an independent poor prognostic factor. We have tried to explain this more fully.

  1. Please briefly discuss in the discussion section alternative i.a. methods of embolization including TAE, TACE and radioembolization (SIRTEX) for liver dominant disease and address their efficacy.

As we are discussing radionuclide treatment we have primarily discussed how our results compare with radioembolization as this is most similar of the alternate liver directed treatments to i.a 90Y DOTATATE. This has been discussed more fully

  1. Please provide a paragraph in the end of the Discussion section with the limitations of the study, including the study design (lack of randomization, no control group) evolving classisfication systems as well as evolving treatment modalities during and after the study period, small sample size etc.

Thank you we agree and we have modified the concluding paragraph to emphasize these issues

Round 2

Reviewer 1 Report

Congratulations on improving your manuscript in such a short period of time. I have no further comments or suggestions